Comparison of chemical compounds associated with sclerites from healthy and diseased sea fan corals (Gorgonia ventalina)

Toledo-Hernández Carlos cgth0918@gmail.com 1
Ruiz-Diaz Claudia P. 1 2
Díaz-Vázquez Liz M. 3
Santiago-Cárdenas Vanessa 4
Rosario-Berrios Derick N. 3
García-Almedina Derek M. 3
Roberson Loretta M. 5
1 Sociedad Ambiente Marino SAM , San Juan , Puerto Rico
2 Department of Environmental Science, University of Puerto Rico , San Juan , Puerto Rico
3 Department of Chemistry, University of Puerto Rico , San Juan , Puerto Rico
4 Department of Biology, University of Puerto Rico , San Juan , Puerto Rico
5 Bell Center, Marine Biological Laboratory , Woods Hole , MA , United States of America
Pawlik Joseph
Electronic publication date: 2017 Aug 25
Publication date: 2017
Volume: 5
Electronic Location ID: e3677
Received 2017 Apr 9; Accepted 2017 Jul 22
Copyright: ©2017 Toledo-Hernández et al.
Copyright year: 2017
Copyright holder: Toledo-Hernández et al.
License: This is an open access article distributed under the terms of the Creative Commons Attribution License, which permits unrestricted use, distribution, reproduction and adaptation in any medium and for any purpose provided that it is properly attributed. For attribution, the original author(s), title, publication source (PeerJ) and either DOI or URL of the article must be cited.
License URL: https://creativecommons.org/licenses/by/4.0/

Keywords: Gorgonia ventalina, Sclerites, Chemical defenses, Aspergillosis

Funding: Puerto Rico Center for Environmental Neuroscience (PRCEN) HRD-1137725 This work was supported by the Puerto Rico Center for Environmental Neuroscience (PRCEN) through an NSF Centers of Research Excellent in Science and Technology (CREST) award, number HRD-1137725. The funders had no role in study design, data collection and analysis, decision to publish, or preparation of the manuscript.

==============================
Background

The roles of gorgonian sclerites as structural components and predator deterrents have been widely studied. Yet their role as barriers against microbes has only recently been investigated, and even less is known about the diversity and roles of the chemical compounds associated with sclerites.

Methods

Here, we examine the semi-volatile organic compound fraction (SVOCs) associated with sclerites from healthy and diseased Gorgonia ventalina sea fan corals to understand their possible role as a stress response or in defense of infection. We also measured the oxidative potential of compounds from diseased and healthy G. ventalina colonies.

Results

The results showed that sclerites harbor a great diversity of SVOCs. Overall, 70 compounds were identified, the majority of which are novel with unknown biological roles. The majority of SVOCs identified exhibit multiple immune-related roles including antimicrobial and radical scavenging functions. The free radical activity assays further confirmed the anti-oxidative potential of some these compounds. The anti-oxidative activity was, nonetheless, similar across sclerites regardless of the health condition of the colony, although sclerites from diseased sea fans display slightly higher anti-oxidative activity than the healthy ones.

Discussion

Sclerites harbor great SVOCs diversity, the majority of which are novel to sea fans or any other corals. Yet the scientific literature consulted showed that the roles of compounds found in sclerites vary from antioxidant to antimicrobial compounds. However, this study fell short in determine the origin of the SVOCs identified, undermining our capacity to determine the biological roles of the SVOCs on sclerites and sea fans.

Introdution

Sea fans (Gorgonia sp.) are one of the top competitors for space within Caribbean coral reefs ecosystems. Similar to their relatives, the scleractinian corals, sea fans have suffered several environmental disturbances such as aspergillosis disease outbreaks (a fungal infection caused by the saprophytic fungus Aspergillus sydowy) that seriously compromised the viability of many populations across the Caribbean during the 1990s (Smith et al., 1996; Nagelkerken et al., 1997; Geiser et al., 1998; Kim & Harvell, 2004). Yet in contrast to the scleractinian, sea fans have overcome these disturbances and continue to thrive. In fact, gorgonians are the prevailing corals in many reefs formerly dominated by scleractinian corals. Their relative success is provided in part, by their strong immune defenses, which have allowed gorgonians to maintain their internal homoeostasis and health (Ruiz-Diaz et al., 2013; Sabat & Toledo-Hernández, 2015).

Gorgonians have a rather diverse repertory of defensive mechanisms to cope and respond to disturbances, both abiotic and biotic (Toledo-Hernández & Ruiz-Díaz, 2014; Ruiz-Diaz et al., 2016). For instance, to respond to pathogen infection, gorgonians are equipped with chemical pathways such as the melanin cascade that prevent or reduce the pathogen dissemination throughout the host tissue (Petes et al., 2003; Mydlarz et al., 2008). Concomitant to this response, gorgonians have the ability to activate the production of peroxidase enzymes, which induce the production of reactive oxygen species with cytotoxic roles and cellular signaling (Mydlarz & Harvell, 2007). Furthermore, gorgonians are also characterized for their diverse production of secondary metabolites, many of which are thought to be immune-related compounds due to their anti-predation and microbial properties (Harvell, Fenical & Greene, 1988; Kim, 1994; Jensen et al., 1996; Smith et al., 1996; Geiser et al., 1998; Kim et al., 2000a; Kim et al., 2000b; Alker et al., 2004).

Gorgonians also have sclerites, which are minute skeletal elements mainly composed of calcium carbonate surrounding an organic matrix made up primarily of glycoproteins, spread throughout the gorgonians’ epidermis, mesoglea, and axial skeleton (Kingsley & Watabe, 1982; Kingsley & Watabe, 1984; Harvell & Suchanek, 1987; Van Alstyne & Paul, 1992). Sclerites are thought to have multiple defensive roles. For instance, sclerites provide structural support to gorgonians by reducing the elasticity and stiffness of the axial skeleton against water motion (Koehl, 1982; Lewis & Wallis, 1991). In addition, sclerites play a role in deterring predation from gastropods and reef fishes (Harvell, Fenical & Greene, 1988; Van Alstyne & Paul, 1992; Koh et al., 2000). Furthermore, the purpling of sclerites (i.e., darkening of sclerites through the accumulation of pigments; Leverette et al., 2008) has also been linked to defensive mechanisms against several biological threats (Petes et al., 2003; Mydlarz & Harvell, 2007; Mydlarz et al., 2008).

Yet the role of sclerites as chemical barriers against microbes has only recently been investigated. For instance, Toledo-Hernández et al. (2016) showed that sea fan sclerites treated with acetone and placed on top of the surfaces of culture media as to block the entrance of microbes, were less effective at delaying the crossing and further colonization of fungi in the agar than sclerites not treated with any solvent. However, no screening of chemical compounds in sclerites was conducted in that study. In fact, studies devoted to surveying the chemical compounds found in gorgonian sclerites are rather scarce, with only a few of them describing the chemical composition of the core of sclerites (Kingsley & Watabe, 1982; Kingsley & Watabe, 1984; Kingsley et al., 1990). Here we perform for the first time, a screening of the semi volatile organic compounds (SVOCs) fraction found in sclerites from healthy and diseased G. ventalina colonies. Furthermore, we performed a scientific literature survey of the SVOCs found in sclerites to determine their potential biological roles. Finally, we conducted radical scavenging activity assays to examine the oxidative potential of chemical compounds associated with sclerites from healthy and diseased fans.

Methods

Tissue collection and sclerite isolation

Sclerites used for this study were isolated from ten healthy and ten diseased Gorgonia ventalina colonies located at a depth of 1.5–2.0 m from El Escambrón beach, San Juan, Puerto Rico (18°28′00″N, 66°05′12″W). The tissue samples were collected under permit 2012-IC-086 issued to Claudia P. Ruiz Diaz, University of Puerto Rico (UPR) Rio Piedras campus, given by the Puerto Rico Department of Natural Resources, Commonwealth of Puerto Rico. Healthy colonies showed neither lesions nor tissue purpling, whereas diseased colonies showed at least one lesion usually overgrown by algae and surrounded by a ring of purple tissue. Previous studies have used the aforementioned pathology to diagnose algal tumor affliction, sea fan aspergillosis, as well as competitive interactions between inter and conspecific individuals (Morse, Morse & Duncan, 1997; Smith et al., 1996; Alker et al., 2004). However, since we performed no microbiological or histological analyses to determine the precise etiology of the injured tissue collected for this study, we avoided the use of any term that could be associated with the afflictions previously described for sea fans. One tissue fragment approximately 4 cm2 was cut from the edge of each healthy fan (hereafter HH). In diseased fans, 2 cm2 samples were cut from both healthy tissue at the edge of the fan (healthy-diseased or HD) as well as from the diseased area (diseased-diseased or DD). Fragments were placed individually in 50 mL centrifuge tubes and brought to the laboratory on ice. Once in the laboratory, each fragment was further cut into smaller fragments of 2–3 cm2 in size and placed in 20.0 mL-sterilized leveled vials filled with 16.0 mL of distilled water. Then, to disassociate sclerites from the soft tissue matrix, vials were vortexed at room temperature, sonicated for 5 min, and then allowed to settle for approximately 4 hrs. This procedure was repeated over a period of 5-7 days to minimize impacts on the structure of the sclerites. After disassociated, sclerites were rinsed several times with distilled water, individually transferred to clean 55 mm Pyrex petri dishes, and dried at 35 °C for 72 hrs.

Gas chromatography-mass spectrometry analysis

From each dried sclerite sample, 0.10 g were grounded in a clean porcelain mortar and the homogenized powder was transferred to a 1.5 mL micro-test tube with 1 mL of acetone. Each tube was vortexed and centrifuged at 6,000 rpm for 5 min. Then, 1.5 mL of the resulting extract from each sample was filtered and transferred without introducing air bubbles to clean amber vials (32 × 12 mm). Afterward, samples were purged with nitrogen to remove dissolve oxygen and then septum caps were fitted to each vial and these were inverted to ensure no headspace was visible and finally, vials were hermetically sealed and not opened until analyzed to preserve the solution. Prior to injecting the solvent used for the extraction, reagents, glassware and other samples processing hardware were blank tested to ensure these materials will not add artifacts to the sclerite extracts analyses. Then, vials with sclerite extracts were placed in an Agilent 7890A gas chromatograph (GC) coupled to an Agilient 5,975 C mass-selective detector for splitless injection. Because removing a sample aliquot from the vials may compromise the integrity of the sample, multiple aliquots were injected to allow for the screening and re-analysis of the samples. The injection temperature was 250 °C. Separation of samples was performed using a fused silica capillary column (SLB®-5 ms 30 m × 0.32 mm × 0.25 μm film thickness; Supelco, Bellefonte, PA, USA). The initial column temperature was held at 70 °C for 1 min, and then increased by 8 °C min−1 until reaching 270 °C, where it was held for 3 min, for a total run time of 29 min per sample. Ultrapure helium at constant-flow of 1.2 mL min1 was used as the carrier gas. Analysis of the generated chromatograms was performed using the AMDIS version 2.7/NIST version 2.0 computer program and the criteria for selection in the analysis for the screening compounds were: (1) R- match > 800; (2) S/N > 15; and (3) peak areas > 1,000. Compound identification was confirmed through the use of NIST library appropriated standards and retention time index (http://www.nist.gov/srd/nist1a.cfm). To determine the potential biological roles of each of the detected compounds, a scientific literature review using the IUPAC names of each compound was conducted.

Free radical scavenging assay

Free radical scavenging activity of sclerite extracts was estimated using 1,1,-diphenyl-2-picrylhydrazyl (DPPH) (MacDonald, Wood & Garg, 2006). Briefly, 0.10 g of homogenized powder not previously used from each of the sclerite samples was placed into labeled and clean 50 mL centrifuge tubes filled with 5.0 mL methanol. Then, each tube was vortexed for 10 s, placed in a water bath at 40 °C for 40 min, and sonicated afterward for 30 min. The samples were then centrifuged for 10 min (6,000 rpm) at 25 °C. Afterwards, 1.0 mL of the supernatant (“sclerite extract”) was transferred to a clean 50 mL centrifugal tube containing 5.0 mL methanol. Then, 1 µL of the sclerite extract and 39 µL of DPPH (0.09 mM) were added to a 96-well microplate and the absorbance of the each sample was recorded at 0, 1, 4 and 5 min and then every 5 min for a one-hour period at 560 nm in a Thermo Scientific Multiscan FC spectrophotometer. Only those samples with absorbances with absorbance <2 were used in the analysis. Scavenging activity in this study was expressed as IC50, which represents the concentration of extracts (mg/mL) needed to inhibit 50% of the DPPH radicals. To calculate IC50 for each sclerite sample, the percent inhibition of radical production (%I) was estimated using the following equation (Mishra, Ojha & Nabo Kumar Chaudhury, 2012): %I=Ai−Af⋅100∕Ai,

where Ai and Af represent the initial and final absorbance, respectively. A linear regression analysis was performed with extract concentration as the independent variable and %I as the dependent variable. Finally, the resulting linear regression equations with a correlation coefficient R2 >80% were used to estimate the IC50.

Statistical analyses

Nonmetric Multi Dimensional Scaling (nMDS) along with an Analysis of Similarity (ANOSIM) with Bray-Curtis distances measurement were utilized to compare the composition of chemical compounds among the sclerites from different tissues. We used nMDS ordination, achieved by the metaMDS wrapper function from the package vegan in R version 3.2. The ordination was applied such that the data was scaled down to two dimensions.

For the analysis of free radical scavenging, a Kruskal-Wallis was performed to detect differences between IC50 index from sclerites from different tissues. In this analysis, IC50 index was assigned as the dependent variable and the origin of the sclerite was assigned as the independent variable. Finally, a Chi2 test was conducted to determine statistical differences in the relative abundances of alcohols, alkanes and alkenes and sclerite origin. In all analyses, P-value was fixed at 0.05. Statistical analyses were performed using R version 3.2 (R Core Team, 2014). For the statistical analyses, HD and DD samples from the same colony where considered independent and analyzed accordingly. In modular organisms such as gorgonian corals, modules are semi-autonomous units. Consequently, any particular stressors (abiotic or biotic) may trigger or have a differential consequence in tissue subjected to different health states (Van Valen, 1978; Tuomi & Vuorisalo, 1989).

Results

Semi-volatile organic compounds

A total of 70 SVOCs were found in the sampled sclerites; 37% of these were identified from DD sclerites, while 35% and 29% were identified from DH and HH sclerites, respectively. Yet the nMDS and ANOMIS analyses failed to group the chemical compounds according to their origin (e.g., HH, HD and DD sclerites). This holds true even when compounds found only once or twice across the samples were eliminated from the analysis.

In general, three major groups were identified: alcohols, alkanes and alkenes. When data was organized based on sclerite origin, alcohols and secondarily alkanes were the most frequently identified SVOCs in all samples, although the number of identified alkenes was higher in HD sclerites (Fig. 1). However, the Chi2 test analysis revealed no statistical differences between alcohols, alkanes and alkenes and the sclerites origin. Furthermore, DD sclerites exhibited the highest number of exclusive compounds with a total of 17 unique SVOCs, followed by HH and HD sclerites with 15 and 8 unique SVOCs, respectively (Table S1 ; Fig. 2). The remaining SVOCs were either shared across all sclerites or between two distinct combinations of sclerites (Fig. 2).

Figure 1 Percentages of the main functional chemicals in sclerites from healthy sea fans (HH), from healthy tissue from diseased sea fans (HD), and diseased tissue from diseased sea fans (DD).

Figure 2 Number of volatile organic compounds (VOCs) in sclerites from healthy sea fans (HH), healthy tissue from diseased sea fans (HD), and diseased tissue from diseased sea fans (DD).

Evidence from the scientific literature concerning the possible biological roles of the SVOCs identified in sclerites was obtained for 44% the compounds. Most were associated with immune related roles such as antimicrobials and antioxidants. Less represented roles were related to cytotoxic and apoptotic compounds, cell membrane components such as steroids and oxidized steroids derivatives, and intermediate signaling molecules.

Figure 3 Average of free radical scavenging analysis expressed as IC50 of sclerites from healthy sea fans (HH), healthy tissue from diseased fans (HD) and diseased tissue from diseased fans (DD).

Bars denote standard errors.

Free radical scavenging assay

Thirteen out of the 30 sclerites sampled tested had R2 >80% and thus were used for the free radical scavenging assays. Of these, six samples were HH sclerites, whereas four and three were DD and HD sclerite samples, respectively. The free radical scavenging IC50 ranged from 0.21 mg dry extract/mL to 0.01 mg dry extract/mL DPPH. Observed differences in antioxidant activity among sclerites were not statistically different, however on average there was a trend for higher values in HH and lowest in DD sclerites (Fig. 3).

Discussion

Overall, 70 SVOCs associated with sclerites were identified. The majority of compounds have not been reported previously in corals and consequently little is known of their biological roles (Table S1). Several of these SVOCs were common to all sclerites samples, whereas others were found in two or exclusively in one of sclerite types (Table S1 ; Fig. 2). Yet no clusters between the compounds and health states were formed, suggesting that the chemical composition of sclerites is similar across the samples independent of the health state of the colony. Nonetheless, the concentration of these compounds may vary among colonies with contrasting health state, but our data do not provide sufficient details to validate or reject this argument.

On the other hand, the scientific literature surveyed revealed potential roles for some of the compounds. For instance, some of the compounds identified in this study have been characterized as free radical scavengers (Table S1). The oxidative analysis assays performed in this study, which revealed that extracts from the three sclerite origins exhibit antioxidant activity, further confirms this fact.

Antioxidant production in gorgonians is thought to have mitigating roles during several immune responses including phagocytosis, peroxidase enzymatic activity, and melanization (Olano & Bigger, 2000; Mydlarz & Harvell, 2007). These immune responses produce reactive oxygen species (ROS) and other free radical molecules, inducing oxidative stress in the coral. Consequently, to counteract this oxidative stress, sea fans have developed regulatory systems such as increased production of non-enzymatic antioxidant scavengers, which neutralize the potential harmful impact of ROS and other free radical molecules (Yost, Jones & Mitchelmore, 2010; Shahbudin et al., 2011).

Antioxidant capacity in corals has also been linked to thermal stress. Several studies have reported that corals under thermal stress exhibit higher antioxidant potential than those under normal temperature conditions (Downs et al., 2002; Griffin & Bhagooli, 2004). In fact, over-expression of genes involved in the oxidative-stress response has been reported to increase in acroporids after they have undergone thermal stress (Császár, Seneca & Van Oppen, 2009). Alternatively, microbes associated with sea fans, both prokaryotes and eukaryotes, may be producing the antioxidant compounds found in sclerites. In the case of eukaryotes such as zooxanthellae for instance, antioxidants may be synthesized as a mitigating mechanism against ROS produced during photosynthesis (Foyer & Shigeoka, 2011).

Compounds known to have antimicrobial properties such as alcohols and esters were also identified in sclerites (Table S1). Many of these compounds have been isolated from terrestrial plants. Nonetheless, the fact that gorgonians produce chemical compounds with antimicrobial activities is not surprising. Multiple culture-based studies using extracts from several gorgonians species have revealed the antimicrobial potential of these extracts (Kim, 1994; Kim et al., 2000a; Kim et al., 2000b; Jensen et al., 1996; Alker, Smith & Kim, 2001; Alker et al., 2004; Ward et al., 2007). Nonetheless, in this study, no relationship could be established between the origin of compounds (i.e., diseased or healthy corals) and their microbial roles, as the majority of the potential roles of the compounds could not be obtained from the scientific literature. Many of these compounds could be precursors or intermediaries of potentially biologically relevant molecules that were trapped within the sclerites during their formation. If indeed some of the compounds isolated from sclerites are precursors of biologically active molecules, as well as environmental cues or internal stressors, sclerites could be used as archives of metabolic pathways or past climatic and biotic stress events.

Conclusions

Sclerites harbor a great SVOC diversity, the majority of which are novel to sea fans or any other corals. However, a major drawback of the study is our inability to determine the origins of these SVOCs. For instance, it is uncertain which SVOCs, if any, were synthetized by microbes associated with sea fans. On the other hand, certain compounds may be derivatives from other compounds. For instance, 1-undercene, 9 methyl, could be a derivative of other compounds such as alcohols which have undergone esterification reactions to form esters that are import fractions of lipids and steroids of living cells. In either case, not knowing the origin of the SVOCs hinder our capacity to understand the underlying mechanism of their biological function. However, as research progresses and our understanding of sclerite formation is improved, coupled with the identification of other chemical compounds using different extraction solvents (polar and non-polar), our understanding of the roles the chemical compounds from sclerites and the corals in general will improve. Yet the present study represents a necessary first step in that process.

Supplemental Information

Table S1 List of semivolatile organic compounds (SVOCs) isolated from sclerites from: healthy fans (HH), from healthy tissue from diseased fans (HD) and diseased tissue (DD) and their respective roles.

Click here for additional data file.

We like to express our gratitude to Alberto M. Sabat for his friendly review.

Additional Information and Declarations

Competing Interests

Author Contributions

Field Study Permissions

Data Availability

The authors declare there are no competing interests.

Carlos Toledo-Hernández and Claudia P. Ruiz-Diaz conceived and designed the experiments, performed the experiments, analyzed the data, wrote the paper, prepared figures and/or tables, reviewed drafts of the paper.

Liz M. Díaz-Vázquez performed the experiments, analyzed the data, contributed reagents/materials/analysis tools, wrote the paper, reviewed drafts of the paper.

Vanessa Santiago-Cárdenas, Derick N. Rosario-Berrios and Derek M. Garcia-Almedina performed the experiments, analyzed the data.

Loretta M. Roberson contributed reagents/materials/analysis tools, wrote the paper, reviewed drafts of the paper.

The following information was supplied relating to field study approvals (i.e., approving body and any reference numbers):

Field experiments were approved by the Department of Natural and Environmental Resources, Commonwealth of Puerto Rico.

The following information was supplied regarding data availability:

The raw data has been supplied as Table S1.

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
