# Peer review of "Comparison of chemical compounds associated with sclerites from healthy and diseased sea fan corals (Gorgonia ventalina)"

_PeerJ, doi:10.7717/peerj.3677_

## Round 0.1 · original submission · Major Revisions

I now have 3 reviews from experts in marine chemical ecology, and all of them have significant concerns about this manuscript in its current form. The authors will have to take great care in preparing a substantial revision that addresses all the reviewers well-considered comments. It is likely that the revision will require additional review.

Reviewer 1 ·

Basic reporting

The manuscript by Toledo-Hernandez and coworkers looks at the effects of disease on the elemental composition and volatile organic compounds content of Gorgonia ventilina sclerites, as well as the antioxidant properties of sclerite extracts. The manuscript was generally clear and unambiguous with professional English being used throughout.

The Introduction referenced relevant literature and generally provided context for the study. It could be improved by presenting specific hypotheses to be tested and the rationale for those hypotheses. For example, it was unclear why the authors measured the elemental composition of the sclerites. Based on background literature, was there a reason to suspect that these would be different?

The manuscript generally generally conforms to the PeerJ standards. Although I did not check all of the references, I noted some were missing from the References section. For example, the references online 74 were absent. The authors should carefully check all of the references they cited to make sure that they are in the References section. They should also make sure that the References Cited are listed alphabetically by the last name of the first author.

Line 201 should reference Table S1.

Table S1 would be easier to follow if the VOCs were put in alphabetical order or were grouped by compound type.

It is not clear why Figure 1 is included in the manuscript. It probably should be deleted.

Figure 2. The authors should write out alcohols instead of using an undefined abbreviation.

Experimental design

Most of the research questions were well-defined and relevant. However, the authors cannot assume that the compounds extracted from diseased Gorgonia were made by the sea fans because pathogens could have been in or on the sclerites of deceased animals and making some of these compounds. It was also not clear why the authors measured the elemental composition of the sclerites. Given that there were no differences between healthy and diseased tissues, the information on elemental analyses could be removed.

The methods had sufficient detail to replicate them.

The statistical analyses were incorrectly done. Because the healthy and disease tissues from diseased Gorgonia were taken from the same individual, they are not independent, violating the assumptions of an ANOVA. Therefore, comparisons should be made between healthy and diseased tissue from the diseased gorgonians with paired t-tests (after checking to see that the data are normally distributed). The authors should then use an unpaired t-test (again, after checking the data for normality) to compare healthy tissues between diseased and healthy individuals. If the authors keep the elemental data in the paper, they need to use multivariate analyses of variance to compare the elements in the tissues. The authors should conduct statistical analyses on the data in Figure 2 to determine whether the percentages are significantly different between tissue types.

Validity of the findings

Lines 220-221, 248-251: The authors should not describe or discuss differences in IC 50 values that are not significantly different.

Lines 223-238: This is introductory material and should be in the Introduction, not the
Discussion.

Discussion: There is an alternative explanation for the patterns seen in the VOCs. It is possible that some of these compounds are being produced by pathogens that are on or have infiltrated the sclerites. This would explain the large number of compounds that are found exclusively in DD (green in Figure 3). There may also be small amounts of these pathogens in the sclerites of healthy tissues of diseased Gorgonia (light red in HD/DD in Figure 3). The pathogens could be using some of these antioxidants to counter ROS that are produced by the gorgonians as a disease response. To conduct these analyses correctly, the authors should have isolated the pathogens and determined the concentrations of VOCs in them.

It appears that the concentration of sclerite extracts used for the DPPH assays was arbitrary. In the discussion, the authors need to note that the results of these assays do not provide information about whether the compounds in the extracts would function as antioxidants in the sclerites (because they were not tested at the actual concentrations in which they occurred). They only provide a means of comparing relative antioxidant activity between tissues.

Reviewer 2 ·

Basic reporting

The paper is fairly clear and well organized. Only some minor grammatical and spelling errors are noted in the annotated pdf. Some mistakes in the works cited should be corrected and the section vetted more carefully.

The study is "self-contained" but falls short of really identifying the role of sclerite bound VOCs and their origin. The former is speculated and the latter is not discussed at all.

Experimental design

Some issues here. There is no detail regarding how the antioxidant assays were conducted with respect to normalizing the concentration of VOCs across samples. If the relative proportion of VOCs is similar between health states but the concentration differs then this could be the reason that antioxidant capacity apparently was lowered in the DD samples.

There was also no indication of how sample alteration was avoided. VOCs are by definitiion volatile so their stability is questionable.

Validity of the findings

There is a lack of reporting on the variation of the data; figs 2 and 3 only report a mean value.

There should be some more advanced analysis (cluster analysis) to determine if individual sample VOC profiles cluster by health status.

Additional comments

Antioxidants are not only used for immunity, they are also very important for mitigating ROS produced by photosynthetic symbionts. In fact, Symbiodinium are quite likely the major source of ROS in the gorgonians relative to immune responses. Consideration of this alternative hypothesis should be added to the discussion and possibly to the introduction as well.

Annotated reviews are not available for download in order to protect the identity of reviewers who chose to remain anonymous.

·

Basic reporting

The manuscript is an interesting new approach to the study of chemistry in gorgonians. It is clear in the general structure, but it can be greatly improved in many aspects. Moreover, it does not completely prove there are no artifacts in the obtained results.
Literature references are ok. Context should be improved (see attached pdf).
Figures and tables are ok. English can be improved in some parts.
The references list of the supplementary table is not correct. Most articles do not include the species name in italics, initial in capitals, and so on. This needs to be corrected.
Overall, it seems to be done in a rush.

Experimental design

The experimental design is ok, as a preliminary approach. However, it would have been much better to include a control sample from a different area. Now, since the results indicate healthy gorgonians do possess all the same antioxidants, you cannot exclude the fact that all the gorgonians in your sampling area are in fact exposed to the pathogens, and therefore, all of them possess the same VOCs. This would be an alternative explanation to your hypothesis saying they may be "prophylactic"...
Another important point is that some of the VOCs can be derivatives of other compounds. Thus, you could be getting artifacts, and this should be deeply discussed, including which measures were taken to ensure the compounds are not artifacts.

Validity of the findings

As said above, the possibility of having artifacts should be clarified. Also, the absence of an external control (from a different area) clearly limits the conclusions that can be drawn from the data. The discussion should include these two facts.

Additional comments

See comments on the pdf and above.

---

## Round 0.2 · accepted · Accept

The authors have addressed the reviewers' comments to a satisfactory degree.

Reviewer 2 ·

Basic reporting

The revised manuscript incorporates suggestions from all reviewers. I have no further comment.

Experimental design

no further comment

Validity of the findings

no further comment

Additional comments

I'm pleased to see the rebuttal and the tracked changes throughout the ms which appears to satisfy most of the reviewer comments.

·

Basic reporting

The revised manuscript is much more clear and rigorous now in its content.
Make sure you correct h (hours) instead of hrs in the final version.

Experimental design

The authors have considerably improved the manuscript in this part.

Validity of the findings

The revised manuscript is more acceptable now with the changes included.

Additional comments

The reviewed manuscript has improved many of the weak points it had in its original form. Now it is much more clear and takes into account the possible artifacts, and includes a more appropriate statistic analysis.